# Exosome-Enhanced Sonodynamic Therapy in Cancer: Emerging Synergies and Modulation of the Tumor Microenvironment

**DOI:** 10.3390/cancers18010118

**Published:** 2025-12-30

**Authors:** Giulia Chiabotto, Marzia Conte, Valentina Cauda

**Affiliations:** 1Candiolo Cancer Institute, FPO-IRCCS, 10060 Turin, Italy; giulia.chiabotto@ircc.it; 2Department of Applied Science and Technology, Politecnico di Torino, 10129 Turin, Italy; marzia.conte@polito.it

**Keywords:** sonodynamic therapy, extracellular vesicles, exosomes, cancer, tumor microenvironment

## Abstract

Standard cancer treatments often cause severe side effects and struggle to reach deep-seated tumors, creating an urgent need for safer, more precise therapies. Sonodynamic therapy (SDT) has emerged as a promising non-invasive approach that uses harmless sound waves to activate special drugs capable of killing cancer cells. However, achieving effective delivery and accumulation of these drugs within tumors remains challenging. This review explores an innovative strategy that employs tiny particles released by cells, called extracellular vesicles (EVs), as smart carriers to deliver these drugs directly to tumors. Since EVs are naturally produced by the body, they are well-tolerated and can be engineered to enhance drug stability and tumor targeting. By combining SDT with EV-based drug delivery, it may be possible not only to destroy cancer cells more precisely, but also to stimulate the immune system to fight the disease, paving the way for safer and more effective cancer therapies.

## 1. Introduction

Cancer remains one of the foremost causes of morbidity and mortality worldwide, accounting for nearly 10 million deaths annually according to the World Health Organization [1]. Despite significant advances in oncology, conventional therapeutic modalities—including surgery, chemotherapy, and radiotherapy—continue to face critical limitations such as invasiveness, systemic toxicity, poor selectivity, and the emergence of multidrug resistance [2]. These challenges underscore the urgent need for novel, non-invasive, and tumor-specific therapeutic strategies that can improve treatment efficacy while minimizing harm to healthy tissues [3].

In this context, sonodynamic therapy (SDT) has emerged as a promising, non-invasive treatment modality that harnesses the unique advantages of low-intensity ultrasound (US) to activate sonosensitizers within tumor tissues [4]. Upon activation, these sonosensitizers generate reactive oxygen species (ROS), highly reactive molecules that damage cellular components, thus triggering apoptosis and other forms of cell death in cancer cells [5,6].

SDT benefits from the deep tissue penetration and spatial precision of US, enabling localized treatment with minimal invasiveness. However, despite its potential, the clinical translation of SDT remains hindered by several challenges, notably those associated with the physicochemical properties of sonosensitizers [7]. These agents often exhibit poor aqueous solubility, rapid systemic clearance, and non-specific biodistribution, leading to suboptimal therapeutic efficacy and unintended off-target effects [8].

To overcome these hurdles, recent research efforts have turned toward leveraging extracellular vesicles (EVs) as natural, biomimetic nanocarriers for the delivery of sonosensitizers. EVs, including exosomes and microvesicles, are nanoscale membrane-bound vesicles secreted by nearly all cell types, playing essential roles in intercellular communication [9]. Their intrinsic attributes—including high biocompatibility, low immunogenicity, and the capacity to traverse biological barriers such as the blood–brain barrier—render them particularly attractive for drug delivery applications [10]. Moreover, EVs can be engineered to carry therapeutic cargos, including sonosensitizers, offering a platform for precise and targeted delivery to tumor sites while minimizing systemic toxicity [11].

The integration of EVs into SDT has recently gained attention as a synergistic strategy that exploits the complementary strengths of both modalities. EVs can enhance the pharmacokinetic profile, stability, and tumor-targeting efficiency of sonosensitizers, while SDT leverages US’s ability to activate these agents deep within tumor tissues with spatial precision [12]. Importantly, this approach holds promise not only for improving direct tumor cell killing but also for modulating the immunosuppressive tumor microenvironment (TME), alleviating hypoxia, and stimulating antitumor immunity [13,14].

Given the growing interest and the translational potential of this innovative therapeutic paradigm, this review aims to provide a comprehensive and up-to-date overview of EV-enhanced SDT (EV-SDT) in cancer treatment, in which natural vesicles released by cells are used to improve the delivery and efficacy of SDT. This review first outlines the fundamental mechanisms of SDT and highlights the unique advantages of EVs as drug delivery systems. It then provides an overview of SDT applications in oncology, with particular attention to the current issues and limitations imposed by the complex tumor microenvironment. A dedicated section focuses on the role of EVs in nanomedicine and cancer, including the main engineering approaches for their functional improvement. Subsequently, the review addresses the strategies for integrating EVs into SDT, and how this combination modulates the TME to overcome existing barriers in cancer therapy. Finally, current challenges and future directions necessary to advance this emerging field toward clinical application are discussed.

## 2. Mechanisms of SDT

SDT consists in the application of a continuous, low-intensity US at diagnostic frequencies, able to deeply penetrate tissues. US waves, commonly used in clinical imaging, are compression and rarefaction mechanical waves that allow the formation, growth, oscillation, and collapse of microbubbles trapped in a fluid. Their explosion can produce high local temperatures and pressures, inducing water thermal dissociation and therefore significant amounts of ROS [15]. In particular, hydroxyl radical (•OH), singlet oxygen (^1^O_2_), and superoxide anion (•O_2_^−^) have been identified as the predominant ROS species responsible for oxidative damage to lipids, proteins, and nucleic acids [16,17]. Other effects include stable cavitation, at lower acoustic pressures, and sonoluminescence [18] due to the violent explosion of the bubbles, which may further contribute to sonosensitizer excitation and ROS generation [19].

Developed on the basis of photodynamic therapy (PDT), which relies on light instead of US to activate photosensitizers, SDT possesses some key distinctive features which make it an incredibly interesting therapeutic option [20]. First, the employed energy source is low-intensity US, able to penetrate much deeper into tissues with respect to light, exceeding 10 cm in some application, without causing heat-related damages [21]. Moreover, US can be focused precisely to a single point, allowing for localized treatment, sparing off-target tissues [22]. Another important feature of US is its inherent safety, allowing for multiple treatment repetitions if necessary [23].

Cavitation affects various biological components: the generation of shockwaves and microjets can mechanically disrupt tissues [15], causing localized increases in temperature [24]; cell membranes can be affected by shear forces and microsteaming, that generate transient pores (sonoporation) [25]; cavitation can cause the transient increases in the permeability of endothelial cell membranes in blood vessels in tumors, facilitating the extravasation of therapeutic agents from the bloodstream [22]; finally, the generation of ROS can hugely impact lipid peroxidation and enzyme activity on a molecular level [21]. Depending on their intracellular concentration, subcellular localization, and duration of exposure, SDT-generated ROS may trigger multiple mechanisms of cancer cell death. Elevated ROS levels primarily induce apoptosis via mitochondrial dysfunction, cytochrome c release, and caspase activation [25,26]. In contrast, moderate oxidative stress may promote autophagic cell death, particularly under hypoxic stress [27,28,29]. In addition, excessive ROS accumulation can drive ferroptosis, a regulated cell death pathway characterized by iron-dependent lipid peroxidation, further expanding the therapeutic mechanisms activated by SDT [27,30,31,32].

SDT combines the application of acoustic waves with the use of sonosensitizing agents, or sonosensitizers [33]. They are acoustically susceptible drug agents or sensitizing molecules, non-toxic on their own but able to elicit cytotoxic effects when activated by US [24]. The predominant mechanism by which sonosensitizer can achieve cytotoxic effects is through the generation of highly reactive ROS. In addition, ideal sonosensitizers should exhibit tumor-targeting capability, biodegradability, and high biocompatibility [34].

Sonosensitizers can be divided into two main categories, namely organic small molecule sonosensitizers and inorganic nanomaterials sonosensitizers [35,36]. Organic sonosensitizers include porphyrins and their derivatives [37], cyanine-based organic sonosensitizers (including IR780 [38]), rose Bengal [39], eosin B [40], and curcumin and chlorophyl [41]; inorganic sonosensitizers can be metal based (TiO_2_ nanostructures [42], manganese-based composites [43], zinc oxide nanocrystals [44,45]) and non-metallic (silicon-based [46], carbon-based [47]). They offer greater stability and superior physiochemical properties compared to the organic ones [48], and can also serve as nanocarriers [49].

To achieve ultimate therapeutic outcomes, it is crucial to efficiently combine advanced equipment—carefully tuning acoustic parameters such as frequency, intensity, and duration of the US [42,50]—and new generations of smart, highly-specific sonosensitizers [22].

## 3. SDT and Cancer Applications

Over the last decades, SDT has been employed as a novel approach for cancer therapy due to its ROS-induced toxicity, which can lead to apoptosis, autophagy, and DNA damage in cancer cells [25,51].

Moreover, it was demonstrated that SDT can resensitize cancer cells to chemotherapy drugs, or it can be combined with other technologies to boost the delivery and effectiveness of chemotherapeutics [52,53], by inducing sonoporation of cancer cells’ membranes or by increasing the permeability of blood vessels surrounding the tumor [54]. Indeed, although SDT was initially explored as a single treatment, it was soon combined with other therapies, creating multimodal approaches that have paved the way for more effective cancer treatments [55]. In this context, the use of US-sensitive nanoparticles(NPs) as drug carriers can help their accumulation in target organs, releasing their cargo and becoming therapeutically effective only when stimulated [53,56].

A serious and recurring issue of a “cold” TME is its hypoxia, which prevents effective drug accumulation [57,58]. SDT was proved to alleviate tumor hypoxia by helping the release of oxygen previously trapped in nanoplatforms [59,60,61,62]. The combination of SDT and photothermal therapy (PTT) created a powerful combination able to overcome the limitations of PTT alone [63,64,65]. SDT was also proved effective in sensitizing tissues to radiation [66], and was employed in combination to radiotherapy in various works, up to the clinical stage [67,68,69]. SDT and immunotherapy were combined and their synergistic effect was documented in different studies [70,71].

When SDT is applied to tumors and ROS production is triggered, a major effect of dying cancer cells is the production of damage-associated molecular patterns (DAMPs), which are cellular molecules normally hidden inside cells but exposed or released during stress or cell death, signaling danger to the immune system. Cancer cells also release tumor-associated antigens that together with DAMPs can activate the immune system and generate an innate immune response [72,73]. Key DAMPs include calreticulin, high mobility group protein 1, adenosine triphosphate, and heat shock proteins (e.g., HSP70, HSP90) [73]. This cascade of events leads to immunogenic cell death (ICD) [25,74], a form of regulated cell death that not only eliminates cancer cells but also primes the immune system to recognize and attack tumor cells, making it a hallmark effect of SDT [75,76]. Indeed, the TME can be drastically changed, and this can hugely impact its nature, switching from a “cold” to a “hot” phenotype [77]. Studies have demonstrated that, when treated with NP-assisted SDT, not only primary tumors reduced their size, but also distant untreated tumors showed a regression due to the establishment of a long-term immune memory able to prevent tumor recurrence [78,79,80].

Therefore, carefully considering the use of multimodal therapies based on SDT to elicit a local therapeutic response while also triggering a long-term, systemic reaction could be the key to design effective novel anticancer treatments [81,82] (Figure 1).

## 4. EVs in Nanomedicine and Cancer

### 4.1. Overview of Exosomes and EVs

EVs are nanoscale, membrane-bound particles released by virtually all cell types into the extracellular space. They encompass different subtypes, mainly exosomes (30–150 nm), microvesicles (100–1000 nm), and apoptotic bodies (>1000 nm), which differ in size, biogenesis, and molecular composition [83]. EVs function as natural carriers of proteins, lipids, metabolites, and nucleic acids, mediating local and systemic intercellular communication. In both physiological and pathological contexts, they play essential roles in immune regulation, tissue repair, and disease progression, including cancer [84]. In fact, EVs released by cancer cells may act both locally, contributing to the onset of a favorable TME, and systemically, by promoting metastatic niche formation [85]. Moreover, their ability to reflect the molecular status of parental tumor cells makes them valuable sources of biomarkers for early disease detection [86].

Exosomes, a well-defined subset of EVs, originate from the endosomal pathway. They form as intraluminal vesicles within multivesicular bodies, which subsequently fuse with the plasma membrane to release exosomes into the extracellular space. These vesicles typically range from 30 to 150 nm in diameter and carry a selective cargo of proteins, lipids, and nucleic acids, such as mRNAs, microRNAs (miRNAs), long non-coding RNAs, and DNA fragments, mirroring the molecular phenotype of their parent cells [87,88].

In cancer, exosomes have emerged as key mediators of tumor progression, actively participating in several hallmark processes, including angiogenesis, metastasis, immune evasion, and remodeling of the TME [89]. Tumor-derived exosomes can promote angiogenesis by transferring pro-angiogenic miRNAs and growth factors, like vascular endothelial growth factor (VEGF), fibroblast growth factor (FGF), and miR-21, to endothelial cells, enhancing vascular sprouting and neovascularization [90]. Moreover, exosomes contribute to metastatic dissemination by facilitating epithelial–mesenchymal transition (EMT), extracellular matrix degradation, and the formation of pre-metastatic niches in distant organs [91]. They also exert immunosuppressive effects, impairing antigen presentation, inhibiting dendritic cell maturation, and modulating T-cell responses, thereby supporting tumor immune escape [92].

Beyond their role as mediators of tumor progression, exosomes are emerging as versatile tools in precision nanomedicine. Thanks to their intrinsic properties—such as high biocompatibility, low immunogenicity, and capacity to traverse biological barriers including the blood–brain barrier—exosomes are increasingly recognized as promising natural nanocarriers for targeted drug delivery in oncology [83,89,93,94]. Their innate targeting capabilities, combined with their ability to encapsulate and protect fragile therapeutic cargos, make them particularly attractive for applications in drug delivery, gene therapy, and immune modulation [95,96,97].

Engineered exosomes can be loaded with chemotherapeutics, small RNAs, or proteins and directed toward tumor cells, improving therapeutic efficacy while minimizing off-target toxicity [98,99]. Moreover, their endogenous origin allows them to navigate complex biological barriers that often limit the performance of synthetic NPs [100]. Collectively, these features underscore exosomes as both biological messengers and therapeutic vectors, bridging the gap between molecular diagnostics and targeted therapy in modern cancer treatment.

### 4.2. Exosome Engineering Strategies for Therapeutic Applications

While native exosomes exhibit considerable therapeutic promise, their direct clinical translation is hindered by several challenges, including heterogeneous composition, low cargo loading efficiency, rapid systemic clearance, and limited targeting specificity [101]. To overcome these limitations, diverse engineering approaches have been developed to tailor exosome function, pharmacokinetics, and tissue tropism. These advances can be broadly classified into three categories: cargo loading, surface functionalization, and exosome mimetics or artificial EVs [93,95,102,103] (Figure 2).

#### 4.2.1. Cargo Loading Techniques

Cargo loading into exosomes can be achieved via endogenous (pre-loading) or exogenous (post-loading) strategies.

Endogenous loading involves engineering donor cells to package therapeutic molecules naturally during vesicle biogenesis. This can be achieved by transfecting parental cells with plasmids encoding therapeutic proteins or with RNA molecules, which can be fused to exosome-associated proteins such as Lamp2b and CD63 to enhance their incorporation into EVs [104]. Alternatively, donor cells can be incubated with small-molecule drugs that become internalized and subsequently packaged into secreted EVs through physiological sorting pathways. This approach ensures biological compatibility and physiological loading but is limited by relatively low yield and complexity in controlling cargo quantity.

Exogenous loading, performed after exosome isolation, encompasses several physical and chemical methods aimed at increasing encapsulation efficiency. Passive loading strategies rely on the intrinsic lipid bilayer of exosomes to encapsulate drugs or small molecules through simple incubation. Hydrophobic compounds such as curcumin [105,106,107], IR780 [108], and paclitaxel (PTX) [109] have been successfully incorporated using this approach, although loading efficiency remains modest [110].

To improve encapsulation efficiency, active loading strategies—such as electroporation, sonication, extrusion, freeze–thaw cycles, and membrane permeabilization with saponin or surfactants —have been developed to facilitate the incorporation of hydrophilic drugs, nucleic acids (siRNA, miRNA), or large biomolecules [111].

Moreover, microfluidic and nanoporation technologies now provide high-throughput, scalable loading with improved control of vesicle size and content uniformity [112,113]. Recent innovations have integrated stimuli-responsive systems, such as pH-sensitive or redox-cleavable linkers, to achieve controlled drug release within TMEs [114]. Collectively, these tools significantly enhance exosome versatility for cancer therapy, enabling delivery of small molecules, nucleic acids, proteins, and imaging probes with improved therapeutic precision.

#### 4.2.2. Surface Functionalization

The biological surface of exosomes offers a platform for engineering targeted delivery, immune evasion, and prolonged circulation. Surface modification strategies are generally divided into genetic engineering and chemical modification [115,116].

Genetic engineering approaches rely on modifying the parental cells to express exosomal membrane proteins fused with targeting ligands or peptides [117]. For example, tumor-homing peptides containing arginine, glycine, and aspartic acid (named internalizing RGD peptides, carrying an RGD sequence), folic acid receptors, monoclonal antibodies [118], or immune checkpoint ligands can be displayed on exosomal surfaces to improve site-specific targeting [119,120]. Membrane proteins enriched on exosomes, such as Lamp2b or the tetraspanin CD63, are commonly used as fusion scaffolds to display targeting ligands on the EV surface [121].

Chemical functionalization, in contrast, can be performed after vesicle isolation, enabling precise and versatile modification without interfering with vesicle biogenesis. Common approaches include carbodiimide-mediated coupling (EDC/NHS chemistry), which activates surface-exposed carboxyl groups on EV membrane proteins or lipids for covalent conjugation with primary amines, allowing efficient attachment of peptides, antibodies, or drug molecules for targeted delivery [116]. Additional approaches include click chemistry reactions (e.g., azide–alkyne cycloaddition) for covalent ligand attachment [122], biotin–streptavidin coupling for high-affinity modular assembly [123], and lipid insertion methods that anchor functional lipids or polyethylene glycol-conjugates directly into the EV membrane to enhance stability and circulation time [124]. These techniques allow flexible post-production modification without altering vesicle biogenesis pathways and aim to improve pharmacokinetics, biodistribution, and therapeutic index of exosome-based delivery systems.

#### 4.2.3. EV Biomimetics and Artificial EVs

To address the limited yield, reproducibility, and standardization issues associated with native exosomes, an active research stream has focused on EV biomimetics—synthetic or semi-synthetic nanovesicles that emulate the structure and function of natural EVs. EV biomimetics are produced through top-down or hybrid strategies that retain biological membrane features while allowing more scalable and tunable production. Parallel to these, fully synthetic artificial EVs aim to replicate key EV functions using bottom-up assembly [125,126,127].

Top-down methods start from whole cells or cell membranes and physically convert them into nanoscale vesicles that retain many native surface markers and membrane proteins. Common techniques include serial extrusion through defined pore membranes, which generates cell-derived nanovesicles, as well as microfluidic shear-force processing or nitrogen cavitation, which fragment the plasma membrane into nanoscale vesicles or membrane pieces that subsequently self-assemble into vesicles [125,128]. These approaches preserve a large fraction of parental membrane proteins, which can mediate homotypic targeting and immune-evasive properties, and deliver relatively high yields compared with naturally secreted exosomes. Top-down vesicles therefore combine biological authenticity with improved production throughput, but they may carry unwanted intracellular components and require rigorous purification and characterization [125,129]. These EV biomimetics can be produced at high yields and preserve many parental membrane proteins which may contribute to immune evasion, tissue homing, and efficient drug delivery for cancer therapy [130,131].

Bottom-up strategies assemble vesicles from defined molecular building blocks, such as lipids, membrane proteins, and polymers, to recreate selected biophysical and biochemical features of natural EVs. This route includes liposome-based formulations with increasingly complex lipid mixtures that mimic natural EV lipidomes, incorporation of purified membrane proteins, and layer-by-layer assembly or templating on synthetic NP cores [126]. Unlike top-down methods, bottom-up assembly enables precise control over vesicle size, composition, surface charge, and stimuli-responsive behavior, allowing the rational tuning of membrane properties such as curvature, polarity, or rigidity to modulate cellular uptake and biodistribution [127,131]. A rapidly expanding direction within this field is the development of fully-synthetic EV-mimicking vesicles designed for both therapeutic and diagnostic applications. Specific studies are indeed introducing strategies to recreate essential EV hallmarks, including characteristic lipid profiles and selected protein motifs, to confer tissue tropism and cargo-delivery capabilities comparable to those of natural EVs [127,131]. By integrating the key functional features of natural EVs with customizable diagnostic or therapeutic capabilities, fully synthetic EV-biomimetics have strong potential for both research and industrial applications. Indeed, these EV-mimicking vesicles offer scalable, reproducible, and on-demand production, along with improved stability that facilitates off-the-shelf storage, and precise control over their morphological and functional properties. Importantly, their defined composition and manufacturing consistency make them more compatible with regulatory and quality standards, ultimately broadening the scope of EV-inspired nanomedicine.

In addition to top-down and bottom-up mimetic strategies, an increasingly explored approach involves generating hybrid vesicles through membrane hybridization [127,129]. This process fuses EV membranes with other biological membranes (e.g., liposomes, immune-cell or erythrocyte membranes). In parallel, polymeric or inorganic nanoparticle cores can be coated with exosomal membranes to form core–shell hybrids, producing chimeric vesicles with enhanced biological and physicochemical properties. Such hybrid platforms benefit from the immune stealth and targeting cues of natural EVs while gaining the versatility, stability, and engineering capacity of synthetic materials. Notably, membrane hybridization has been shown to improve immune evasion, prolong circulation, and increase adaptability within the tumor microenvironment, making these systems particularly promising for advanced cancer therapies.

Hybrid and core–shell designs bridge top-down and bottom-up benefits. These systems typically include membrane-coated NPs, where polymeric, inorganic, or lipid-based cores (e.g., poly(lactic-co-glycolic acid), silica, or metal oxide) are cloaked with cell- or exosome-derived membranes [132]. By combining the tunable physicochemical properties of the NP core (e.g., controlled size, high drug loading, stimuli responsiveness) with the biocompatible, immune-stealth membrane derived from exosomes, these biomimetic NPs achieve superior payload capacity, targeting specificity, and circulation time [133,134].

Overall, EV biomimetics and hybrid EVs represent promising platforms for translating EV-based functions into manufacturable nanomedicines that balance biological performance with industrial feasibility [135].

### 4.3. Exosomes in the TME

Beyond serving as nanocarriers for therapeutic cargos, exosomes play an active role in shaping and remodeling the TME [136]. Their ability to transfer proteins, lipids, and nucleic acids enables them to regulate immune responses, metabolic activity, angiogenesis, and metastatic progression, making them key modulators of tumor biology and important tools for therapeutic intervention.

Tumors exploit exosomes to establish an immunosuppressive milieu by transporting immune checkpoint ligands (e.g., programmed death-ligand 1) or immunosuppressive cytokines [137]. Engineered exosomes, however, can reverse this suppression by delivering immunostimulatory cargos such as tumor antigens, adjuvants, or checkpoint inhibitors [138,139]. For example, dendritic cell-derived exosomes carrying MHC-peptide complexes and costimulatory molecules have been shown to stimulate T-cell activation and induce durable antitumor immunity [140].

Cancer cells display metabolic plasticity to sustain rapid growth in nutrient-deprived environments. Exosomes can be engineered to disrupt these metabolic pathways. EV-based strategies targeting mitochondrial metabolism or hypoxia adaptation are under investigation to weaken tumor resilience and improve therapy responsiveness [141].

Exosomes derived from tumor cells often promote angiogenesis by transporting pro-angiogenic factors (VEGF, FGF, angiopoietins) and microRNAs such as miR-210 and miR-21-5p, facilitating neovascularization and metastatic spread [142,143,144]. Conversely, therapeutic exosomes can be loaded with anti-angiogenic agents or miRNAs (e.g., miR-9, miR-29b) to suppress tumor vasculature development [145,146]. Moreover, exosomes influence EMT and pre-metastatic niche formation, processes that can be therapeutically targeted to prevent dissemination [147].

## 5. Integration of EVs in SDT

The integration of EVs—particularly exosomes—into SDT has emerged as an innovative approach to overcome the limitations of conventional sonosensitizers, including poor solubility, low tumor selectivity, and rapid systemic clearance. Beyond serving as biocompatible and intrinsically targeted drug carriers, EVs actively participate in the mechanistic cascade underlying SDT. Upon US irradiation, EV-delivered sonosensitizers benefit from enhanced cellular internalization and spatial confinement within tumor tissues, resulting in amplified and localized ROS generation. US-induced phenomena such as acoustic cavitation and sonoporation further increase membrane permeability, facilitating EV uptake and intracellular payload release.

Recent mechanistic insights indicate that the interplay between US stimulation and EV-mediated delivery not only enhances sonosensitizer activation but also promotes mitochondrial dysfunction, oxidative stress amplification, and ICD pathways, thereby strengthening the therapeutic outcome [148]. In parallel, engineered exosomes actively remodel the TME by modulating immune responses, altering tumor metabolism, and suppressing angiogenesis, effectively alleviating key barriers such as hypoxia and immune suppression that commonly limit SDT efficacy. These multifunctional properties enable engineered exosomes to synergize with SDT not only by enhancing tumor-specific accumulation and improving the activation of therapeutic payloads, but also by mitigating key microenvironmental barriers, including hypoxia and immune suppression, that often hinder treatment efficacy [30]. Collectively, this combined approach offers a powerful platform to increase the precision, potency, and therapeutic durability of SDT-based cancer treatment [107].

Exosomes can encapsulate or be hybridized with sonosensitizers to improve their stability and tumor accumulation. Dumontel et al. [149] developed a ‘TrojanNanoHorse’ by re-engineering B-cell-derived EVs with zinc oxide nanocrystals via freeze-thaw loading and decorating their surface with anti-CD20 monoclonal antibodies. This system showed high targeting specificity and enhanced on-demand cytotoxicity against CD20^+^ lymphoma cells (Daudi) when activated by US shock waves. Similarly, Nguyen Cao et al. [150] developed biocompatible exosome-based nanosonosensitizers carrying IR780, achieving tumor-specific accumulation and efficient US activation. Further refinements include stimuli-responsive and multifunctional systems, such as the dual stimuli-sensitive exosomes described by Nguyen Cao et al. [151], which combined chemo-SDT with photoacoustic imaging for spatiotemporal control of drug release.

To improve tumor homing, surface-engineered and biomimetic exosome formulations have been proposed. Wu et al. [152] designed macrophage exosome-disguised nanoplatforms that could cross the blood–brain barrier and deliver sonosensitizers to glioblastoma (GBM), markedly enhancing therapeutic outcomes. Similarly, Li et al. [153] utilized tumor-derived exosomes to deliver curcumin for colon cancer SDT, where US-triggered calcium overload amplified cell death.

As already mentioned, the achievement of multimodal therapies relying not only on the enhancement of the SDT therapeutic effect by means of EVs but also involving other types of external stimulation, such as PDT, has also been the object of study. Nguyen and colleagues implemented biotin-conjugated exosomes containing a backbone of the sonosensitizer IR820, further enriched by a heavy atom-free sensitizer to achieve fluorescence imaging-guided sono-PDT upon NIR light/US irradiation [154] (Figure 3).

These bioinspired platforms demonstrate how exosomal membranes confer selective tropism, immune evasion, and improved pharmacokinetics compared to synthetic NPs (Table 1).

## 6. Modulating the TME via EVs-SDT

Beyond drug delivery, exosome-based SDT systems can remodel the TME by alleviating hypoxia, reprogramming metabolism, or eliciting antitumor immunity. As summarized in Figure 4, precise engineering of EVs, combined with SDT, can transform the TME from an immunologically “cold” state into a “hot” one, promoting immune activation. This approach enhances tumor oxygenation, increases ROS generation, and stimulates cytotoxic immune cell recruitment and activity. Collectively, these effects reduce immunosuppressive signals and modulate tumor-stroma interactions, creating conditions that support more effective and durable antitumor responses.

Several studies have explored the use of EV-based SDT to modulate the TME through complementary mechanisms. In order to enhance oxidative and metabolic stress, Nguyen Cao et al. [155] developed mitochondria-targeting exosomes to amplify ROS-mediated apoptosis. As further refinements, they introduced bioreducible exosomes encapsulating glycolysis inhibitors, promoting metabolic collapse and enhanced oxidative stress [156].

The reversion of an immunosuppressive condition coupled with the activation of an antitumor immunity has also been pursued. Wang et al. [157] further demonstrated that engineered exosomes could reverse immunosuppression in prostate cancer, promoting dendritic cell activation and cytotoxic T-cell infiltration.

Targeting cancer-derived exosome biogenesis represents another possible distinct approach. Wu et al. [158] employed a polymeric sonosensitizer to selectively inhibit tumor-derived exosome biogenesis and release. Since cancer-secreted exosomes contribute to immune evasion and therapy resistance, this strategy disrupted oncogenic intercellular signaling while simultaneously promoting SDT. The resulting system achieved tumor-specific immune activation, reduced metastatic communication, and enhanced the efficacy of SDT. Similarly, Qu and colleagues developed nanocomposites made of manganese-doped hydroxyapatite modified with an ROS-cleavable lipid and loaded with an exosome inhibitor. The application of US induced the cleavage of the lipid, efficiently releasing the exosome inhibitor, which in turn triggered an anticancer immune response and delayed tumor growth [159].

Finally, overcoming tumor hypoxia remains a major limitation for SDT, reducing ROS yield. To address this, Wu et al. [30] engineered oxygen-boosted exosomes integrating CAT-mimetic NPs to decompose endogenous H_2_O_2_, generating O_2_ and enhancing ROS formation. This system also induced ferroptosis, expanding SDT’s therapeutic mechanisms beyond apoptosis (Figure 5).

Collectively, these studies underscore the potential of exosome-integrated SDT to combine biological specificity with physical precision, achieving superior therapeutic outcomes through targeted ROS production, immune activation, and metabolic reprogramming (Table 2).

## 7. Challenges and Clinical Translation

Although promising in preclinical settings, all the above-mentioned nanotechnologies employing EVs as engineered nanocarriers for enhanced SDT still face major barriers to clinical translation. These challenges largely mirror those encountered in standalone EV-based applications, including low yield and heterogeneity of EV preparations, lack of standardized isolation and purification methods, and limited scalability of production under Good Manufacturing Practice (GMP) conditions [160,161,162]. In addition, batch-to-batch variability, differences in EV composition depending on the cell source and culture conditions, and the absence of universally accepted characterization criteria complicate reproducibility and regulatory approval [163].

Another key limitation lies in insufficient targeting precision and biodistribution control in vivo. Despite their inherent biocompatibility and tropism, EVs often accumulate in off-target organs such as the liver and spleen due to uptake by the mononuclear phagocyte system, which reduces their therapeutic index [164]. Furthermore, uncertainties regarding pharmacokinetics, biodistribution, and clearance mechanisms raise safety concerns that must be addressed before clinical application [165].

Finally, regulatory and ethical hurdles remain substantial. Defining EVs as biological or nanotechnological products impacts the approval route, quality control requirements, and intellectual property management. Establishing robust large-scale biomanufacturing, storage stability, and long-term safety data is therefore essential to enable clinical-grade EV formulations for SDT and other therapeutic modalities [96,166].

The incorporation of sonosensitizers, inorganic NPs, metal moieties, drugs, and immune adjuvants, creates even more intricate regulatory issues and obstacles to clinical translation. Indeed, most inorganic nanomaterials are nondegradable and contain metal ions, whose accumulation may lead to systemic toxicity. Moreover, the combination of multiple active agents in EVs may lead to significant challenges to clinical dosing, and the engineering techniques used in many of the reported studies complicate the path towards mass production under GMP standards. Additionally, although the benefits of SDT treatments such as inherent safety of the irradiation source and non-invasiveness have been thoroughly highlighted, their clinical translation is hindered by the poor overall response to such therapies. In fact, the TME poses important obstacles to the efficacy of SDT such as a high pH, a high interstitial fluid pressure, and hypoxia, which significantly limit ROS yield [81]. Finally, active targeting is pivotal for ensuring efficient tumor accumulation, necessary to elicit a response resulting from the US irradiation. Most of the studies reported here have, however, partially addressed some of these issues, efficiently targeting the tumor mass and resulting in an effective shift from cold to hot TME and important immune recruitment [155,157].

The most commonly employed strategy to achieve good treatment efficacy is a combination of therapies including SDT as adjuvant [167]. Indeed, current clinical trials involving SDT have only focused on its use as adjuvant therapy, therefore in combination with chemotherapy and radiotherapy, and mainly with glioma patients (Table 3).

Recently, a phase I clinical trial exploring the combination of SDT and radiotherapy following hematoporphyrin administration was conducted and completed on 11 patients with brainstem gliomas, resulting in a well-tolerated combination therapy and a partial response in 2 of them [69]. Currently, a clinical trial (NCT06039709) is evaluating the combination of 5-Aminolevulinic acid (5-ALA, Gleolan) and neuronavigation-guided low-intensity focused US on patients with recurrent GBM [168]. A phase II study (NCT05123534) three years ago examined the safety, pharmacokinetics, and preliminary efficacy of ascending drug and energy dose combinations for SDT using SONALA-001 in combination with Exablate 4000 type 2.0 MR-Guided Focused US on subjects with diffuse intrinsic pontine glioma and diffuse midline glioma, but results have not been published yet [169]. Another phase II study (NCT07130149) is currently enrolling patients to test the combination of SDT (Hiporfin drug administration and focused US sessions) with standard therapy (radiation, chemotherapy, bevacizumab) in patients with GBM [170]. Finally, another phase I trial (NCT06665724) is currently recruiting patients to evaluate the 5-ALA combined with CV01 delivery of US for SDT in patients with newly diagnosed high-grade glioma prior to resection and standard adjuvant therapy [171].

Collectively, these limitations highlight the need for standardized protocols, scalable production technologies, and comprehensive preclinical-to-clinical pipelines to realize the translational potential of EV-engineered sonodynamic therapies. We are therefore confident that the promising results obtained in the studies reported in this review will eventually pave the way for novel clinical trials based on solid preclinical evidence combining SDT and engineered EVs, able to achieve a superior therapeutic efficacy.

## 8. Conclusions

The integration of EVs with SDT has proven to be a highly synergistic and promising treatment in cancer nanomedicine. The preclinical evidence here presented shows that EVs-SDT can effectively overcome the core limitations of standalone sonosensitizers, namely poor solubility and non-specific biodistribution. More significantly, this combined therapy was proven capable of achieving a reprogramming of the TME, alleviating hypoxia and promoting a shift from an immune-suppressive “cold” to an immune-activated “hot” TME.

Despite such promising achievements, the path to clinical translation faces critical bottlenecks, including the regulatory complexity associated with multi-cargo systems, potential long-term safety concerns and, most importantly, challenges related to the scalability and standardization of EVs manufacturing.

Nevertheless, the increased presence of clinical trials involving the use of SDT as a therapy adjuvant validates the therapeutic potential of US-activated treatments, and we are confident that rigorous preclinical development, focused on overcoming the above-mentioned limitations, will pave the way for novel clinical trials based on EV-SDT combined therapy.

## Figures and Tables

**Figure 1 cancers-18-00118-f001:**
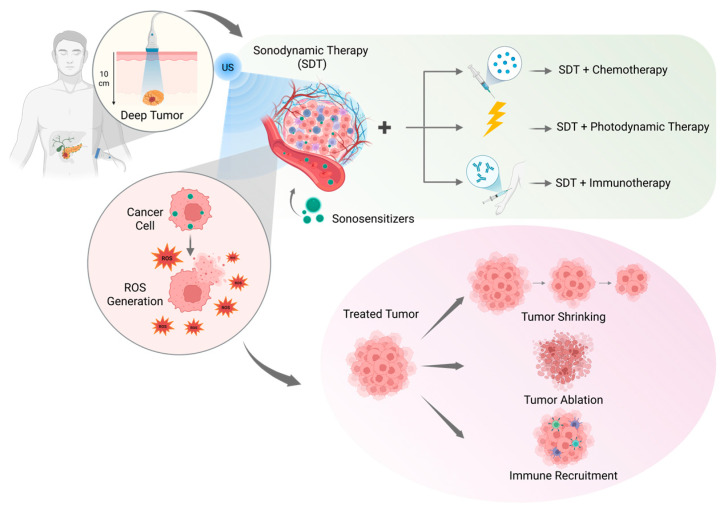
Scheme reporting the application of SDT for the treatment of deep-seated tumors; its combination with chemotherapy, PDT, and immunotherapy to obtain robust multimodal therapies; the ROS generation caused in cancer cells; and the possible resulting effects on the tumor mass. Created with Biorender.com.

**Figure 2 cancers-18-00118-f002:**
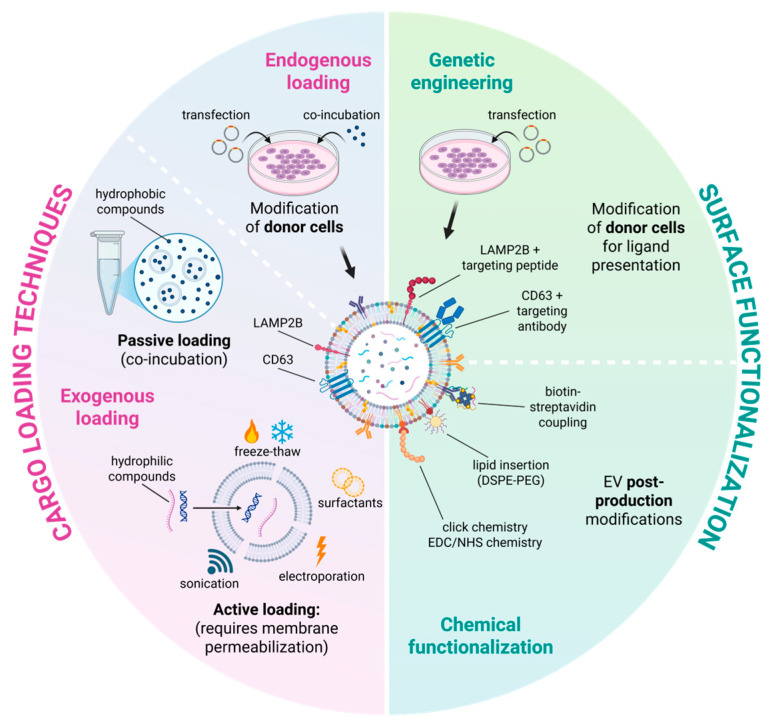
EV engineering strategies for cargo loading and surface functionalization. Schematic representation of the major approaches used to engineer EVs as cancer therapeutics. Cargo loading can be achieved through endogenous strategies—where donor cells are modified to package therapeutic molecules during EV biogenesis—or through exogenous strategies applied after EV isolation. Surface functionalization methods include genetic engineering of donor cells to display targeting ligands on secreted EVs, as well as post-isolation chemical modification approaches. These complementary strategies enable enhanced targeting, improved stability, and tailored therapeutic performance of engineered EVs. Created with Biorender.com.

**Figure 3 cancers-18-00118-f003:**
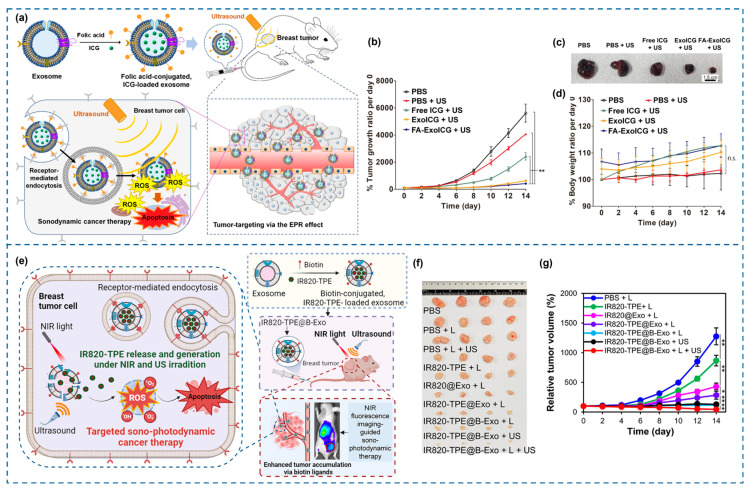
(**a**) Scheme reporting the fabrication process and the SDT-based therapeutic action of folic acid (FA)-conjugated, indocyanine green (ICG)-loaded exosomes (FA-ExoICG): once injected, they selectively accumulate in the tumor and are internalized by folate receptor overexpressing breast cancer cells (MCF-7), producing cytotoxic ROS in response to US, leading to apoptosis. (**b**) Normalized tumor growth ratio of MCF-7 tumor xenografts in treated vs. untreated groups. ** *p* < 0.01. (**c**) Representative digital photo of explanted tumors. (**d**) Body weights of mice over time. n.s.: not significant. Adapted with permission from [150]. (**e**) Schematic illustration of the sono-photodynamic therapy employing IR820-TPE@B-Exo (4-(1,2,2-triphenylvinyl)-1,1′-biphenyl (TPE) conjugated to the IR820 sonosensitizer backbone loaded into biotin-conjugates exosome) for breast tumor-targeted therapy upon NIR light (L) and US irradiation. (**f**) Digital photos of the extracted tumors. (**g**) Tumor volume progression over time comparing the various treatment groups. * *p* < 0.05, ** *p* < 0.01. Adapted with permission from [154].

**Figure 4 cancers-18-00118-f004:**
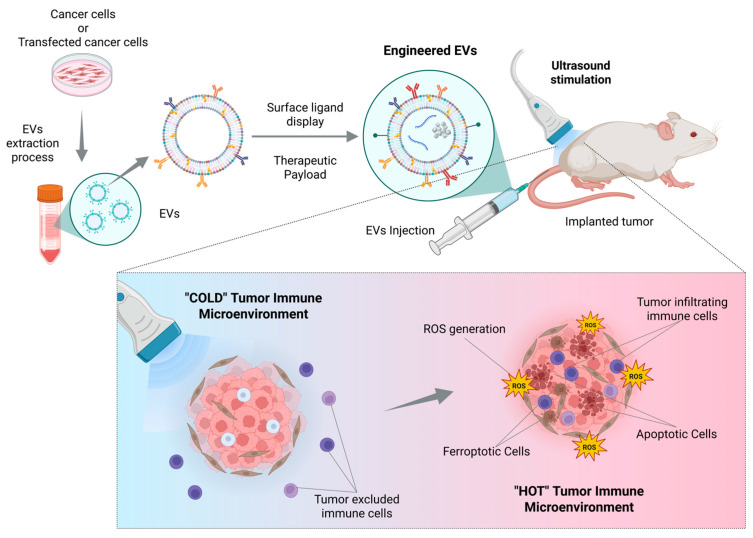
Scheme illustrating the effects of EV-SDT on the tumor and the surrounding microenvironment. Precise engineering of the EVs, coupled with SDT, results in a switch from a “cold” to a “hot” TME. Created with Biorender.com.

**Figure 5 cancers-18-00118-f005:**
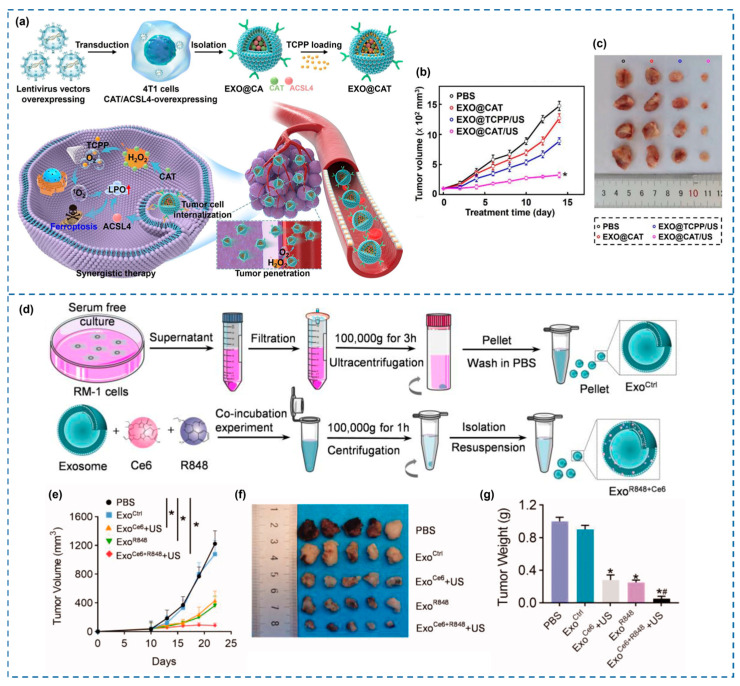
(**a**) Schematic illustration of the preparation EXO@CAT NVs and treatment mechanism based on tumor vessel extravasation, tumor penetration, and cell internalization, followed by SDT and ferroptosis. (**b**) Tumor growth curves among all treatment groups. * *p* < 0.05. (**c**) Representative pictures of explanted tumors. Reproduced with permission from [30]. (**d**) Scheme of the fabrication of Exo^Ce6þR848^ and control counterpart. (**e**) Mean tumor growth curve of primary tumors subjected to the treatments. * *p* < 0.05. (**f**) Digital photo of the explanted tumors. (**g**) Weights of the explanted tumors. * *p* < 0.05, # *p* < 0.05. Reproduced with permission from [157].

**Table 1 cancers-18-00118-t001:** Representative works focusing on the integration of EVs in SDT, and illustrative therapeutic outcomes of the combined therapies.

EVs Origin	Engineering Strategy	Target Tumor Model	Sonodynamic Treatment	Therapeutic Outcome	Ref.
Lymphocyte cell line (IST-EBV-TW6B)	**Sensitizer:** ZnO NCs; **Ligand:** anti-CD20; **Method:** Freeze-thaw loading and surface modification.	Burkitt’s lymphoma Daudi cells (CCL213)	**Shock waves**: 250 shots of 12.5 MPa, 4 shots/s, 3 times/day, every 3 h (in vitro)	**Effect:** Selective CD20^+^ cytotoxicity; **Outcome:** Significant on-demand cancer cell death.	[149]
Human embryonic kidney HEK-293T cells	**Sensitizer:** ICG; **Ligand:** Folic acid; **Method:** Surface functionalization and cargo loading.	Breast cancer cells (MCF-7) and human dermal fibroblasts (h-DFB) in vitro; MCF-7 xenografts in vivo	**US**: 0.3 W/cm^2^, 1 MHz, 60/70 s; 0.5 W/cm^2^, 1 MHz, 3 min	**Effect:** Selective tumor accumulation; **Outcome:** Effective suppression of tumor growth.	[150]
Human embryonic kidney HEK-293T cells	**Sensitizer:** ICG; **Cargo:** PTX + sodium bicarbonate; **Method:** Combined nanocarrier encapsulation.	Breast cancer cells (MCF-7) in vitro, MCF-7 xenografts in vivo	**US**: 0.3 W/cm^2^, 1 MHz, 1 min (in vitro) 0.5 W/cm^2^, 1 MHz, 3 min	**Effect:** Facilitated cytoplasmic drug release; **Outcome:** Significant inhibition of proliferation.	[151]
Human reticulum cell sarcoma cells (J774A.1)	**Cargo:** CAT-Silica NPs; **Ligand:** AS1411 aptamer; **Method:** Aptamer-decorated exosome embedding.	Human keratinocyte (HaCat), endothelial (bEnd 3) and GBM (U87) cells (in vitro); luciferase-expressing U87 tumors (in vivo)	**US**: 1.5 W/cm^2^, 1 MHz, 40% duty cycle (in vivo)	**Effect:** Growth inhibition; **Outcome:** Reduced metastasis and primary tumor volume.	[152]
Murine colon adenocarcinoma cells (CT26)	**Sensitizer:** Curcumin; **Cargo:** Calcium carbonate NPs; **Method:** Exosomal co-encapsulation.	Colon adenocarcinoma cells (CT26) in vitro; CT26 subcutaneous tumor model (in vivo)	**US**: 1.5 W/cm^2^, 1 MHz, 30% duty cycle, 10 min (in vivo)	**Effect:** pH-responsive/SDT synergy; **Outcome:** Robust tumor suppressor effect	[153]
Human embryonic kidney HEK-293T cells	**Sensitizer:** 4-(1,2,2-triphenylvinyl)-1,1′-biphenyl (TPE)-IR820; **Ligand:** Biotin; **Method:** Loading into biotin-conjugated exosomes	Human breast cancer cells (MCF-7) in vitro; MCF-7 xenografts in vivo	**US**: 0.5 W/cm^2^, 1 MHz, 2 min (in vitro and in vivo)	**Effect:** Sono-PDT synergy; **Outcome:** Maximized tumor targeting and growth inhibition.	[154]

**Table 2 cancers-18-00118-t002:** Examples of TME modulation, achieved through sophisticated engineering of the EVs to enhance the effects of SDT.

EVs Origin	Engineering Strategy	Target Tumor Model	Sonodynamic Treatment	TME Modulation	Ref
Human embryonic kidney HEK-293T cells	**Sensitizer:** Ce6-TPP (mitochondria-targeted); **Cargo:** Piperlongumine; **Method:** Co-encapsulation.	Human breast cancer cells (MCF-7) and human dermal fibroblasts (h-DFB) in vitro; MCF-7 xenografts in vivo	**US**: 0.3 W/cm^2^, 1 MHz, 1 min (in vitro) 0.3 W/cm^2^, 1 MHz, 3 min (in vivo)	**Mitochondrial oxidative stress**: selective and maximal cytotoxic destruction of cancer cells	[155]
Human embryonic kidney HEK-293T cells	**Sensitizer:** Ce6-TPP; **Cargo:** Glycolysis inhibitors; **Lipids:** GSH-responsive diselenide-bearing	Human breast cancer cells (MCF-7) in vitro; MCF-7 xenografts in vivo	**US**: 0.5 W/cm^2^, 1 MHz, 2 min (in vitro) 0.5 W/cm^2^, 1 MHz, 3 min (in vivo)	**Glycolysis inhibition**: energy depletion and oxidative stress-induced tumor death	[156]
Human embryonic kidney HEK-293T cells	**Sensitizer:** Ce6; **Adjuvant:** R848 (TLR7/8 agonist); **Method:** Cargo loading.	Murine dendritic cells (DC2.4), murine macrophage cells (RAW 264.7) in vitro; murine embryonic prostate cells (RM-1) tumors in vivo	**US**: 0.1/0.5/1 W/cm^2^, 1 MHz, 20% duty cycle, 5 min (in vitro) 2 W/cm^2^, 1 MHz, 20% duty cycle, 5 min (in vivo)	**Immune remodeling (Cold-to-Hot)**: macrophage repolarization, DC maturation, and T-cell activation.	[157]
Transfected murine breast cancer cells (4T1)	**Sensitizer:** TCPP (tetrakis (4-carboxyphenyl) porphyrin); **EV Modification:** CAT/Acyl-CoA enrichment via transfection.	Murine breast cancer cells (4T1) in vitro;	**US**: 1.5 W/cm^2^, 1 MHz, 3 min (in vitro and in vivo)	**Hypoxia alleviation:** O_2_ production-driven nanoconstruct motion; metastatic modulation	[30]

**Table 3 cancers-18-00118-t003:** Current ongoing and completed clinical trials including SDT as adjuvant.

ID Trial	Tumor Type	Drug Used with SDT	Enrolled Patients	Ref.
ChiCTR2200065992	Brainstem gliomas	Hematoporphyrin (Xipofen)	11	[69]
NCT06039709	GBM	(5-ALA)	11 (est.)	[168]
NCT05123534	Diffuse Intrinsic Pontine Glioma and Diffuse Midline Glioma	Proprietary intravenous formulation of 5-ALA (SONALA-001)	15	[169]
NCT07130149	GBM	Hiporfirin^®^	230 (est.)	[170]
NCT06665724	Recurrent High Grade Glioma	5-aminolevulinic acid hydrochloride (5-ALA HCl, Gliolan^®^)	14 (est.)	[171]

## Data Availability

No new data were generated or analyzed in this study. Data sharing is not applicable to this article.

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
