# Peer review of "Exosome-Enhanced Sonodynamic Therapy in Cancer: Emerging Synergies and Modulation of the Tumor Microenvironment"

_cancers, 2025, doi:10.3390/cancers18010118_

Round 1
Reviewer 1 Report
Comments and Suggestions for Authors
This review focuses on the application of exosome-enhanced sonodynamic therapy (EV-SDT) in cancer treatment, systematically summarizing the mechanisms of sonodynamic therapy (SDT), the nanomedical properties of extracellular vesicles (EVs), the strategies for their integration, the regulatory effects on the tumor microenvironment (TME), as well as the challenges and future directions in clinical translation. The topic closely aligns with the cutting-edge needs of cancer therapy, centering on the core hotspot of non-invasive precision treatment. The review is comprehensive in content, logically structured, and cites novel and extensive literature, providing a valuable reference framework for researchers in this field with high academic value and application prospects. However, several issues need to be addressed prior to formal publication.
- The manuscript contains numerous abbreviations (e.g., EV-SDT, DAMPs, ICD). Although abbreviations are explained at the end, brief popular explanations should be added when each abbreviation first appears to lower the comprehension barrier for non-professional readers. For some complex mechanisms (such as the synergistic effect between ferroptosis and SDT), a concise logical chain explanation can be supplemented to make the expression more understandable.
- The abstract is excessively descriptive. It is recommended to first highlight the existing research gap—for example, the lack of comprehensive reviews focusing on exosome-enhanced sonodynamic therapy—and then introduce the core content of this manuscript, clarifying the novelty and value of the review.
- At the end of the introduction, it is suggested to summarize the main content of the subsequent manuscript, clearly informing readers of the key focus of this review (e.g., the integration strategies of EVs and SDT, the regulatory mechanisms of TME, and the latest progress in clinical translation), to help readers quickly grasp the overall structure and core value of the article.
- The section on the mechanisms of sonodynamic therapy and exosomes is well-written. It is recommended to refer to the expression logic and technical details in the following article to further optimize the clarity and depth of the mechanism description: Nano Research Volume 16, pages 782–791, (2023). This will help to more accurately and comprehensively present the interaction mechanisms between EVs and SDT.
- Some tables in the manuscript (e.g., Table 2) suffer from incomplete content layout and missing information, and the table details should be supplemented and improved. In addition, a small number of sentences have poor grammatical coherence (e.g., the expressions related to "reversing immunosuppression" in the abstract), and the Chinese expression needs to be optimized to enhance the fluency of the text.
The English could be improved to more clearly express the research.
Author Response
This review focuses on the application of exosome-enhanced sonodynamic therapy (EV-SDT) in cancer treatment, systematically summarizing the mechanisms of sonodynamic therapy (SDT), the nanomedical properties of extracellular vesicles (EVs), the strategies for their integration, the regulatory effects on the tumor microenvironment (TME), as well as the challenges and future directions in clinical translation. The topic closely aligns with the cutting-edge needs of cancer therapy, centering on the core hotspot of non-invasive precision treatment. The review is comprehensive in content, logically structured, and cites novel and extensive literature, providing a valuable reference framework for researchers in this field with high academic value and application prospects. However, several issues need to be addressed prior to formal publication.
We thank the Reviewer for their kind comments on our manuscript. We have addressed all the issues raised in the following answers.
- The manuscript contains numerous abbreviations (e.g., EV-SDT, DAMPs, ICD). Although abbreviations are explained at the end, brief popular explanations should be added when each abbreviation first appears to lower the comprehension barrier for non-professional readers. For some complex mechanisms (such as the synergistic effect between ferroptosis and SDT), a concise logical chain explanation can be supplemented to make the expression more understandable.
We thank the Reviewer for this suggestion. As the Reviewer highlighted, we have already provided a thorough list of abbreviations and a simple summary specifically addressing non-professional readers. To improve readability for non-specialist readers, we have revised the manuscript by introducing brief and intuitive explanations of the following abbreviations at their first appearance: ROS (page 2 lines 64-65), EV-SDT (page 2, lines 90-91), PDT (page 3 lines 114-115), DAMPs (page 4 lines 178-179), ICD (page 4, lines 183-185). In addition, we have clarified complex mechanistic processes by adding concise logical descriptions where appropriate. In particular, in Paragraph 2 the section describing SDT-induced cell death pathways has been expanded to explicitly explain how different levels and localization of ROS contribute to apoptosis, autophagy, and ferroptosis (page 3, lines 128-136). These modifications aim to lower the comprehension barrier while preserving scientific accuracy.
2.The abstract is excessively descriptive. It is recommended to first highlight the existing research gap—for example, the lack of comprehensive reviews focusing on exosome-enhanced sonodynamic therapy—and then introduce the core content of this manuscript, clarifying the novelty and value of the review.
We thank the Reviewer for her/his suggestion. We have modified the Abstract accordingly (page 1, lines 38-40).
3. At the end of the introduction, it is suggested to summarize the main content of the subsequent manuscript, clearly informing readers of the key focus of this review (e.g., the integration strategies of EVs and SDT, the regulatory mechanisms of TME, and the latest progress in clinical translation), to help readers quickly grasp the overall structure and core value of the article.
We thank the Reviewer for this constructive suggestion. The final paragraph of the Introduction has been revised to provide a clear and structured overview of the manuscript (see pages 2-3, lines 89-101). Specifically, we now explicitly outline the key topics addressed in the review, including (1) the fundamental mechanisms of SDT, (2) the role and engineering of EVs in cancer, (3) strategies for integrating EVs with SDT, (4) the modulation of the tumor microenvironment, and (5) current challenges and future directions for clinical translation. We believe that this revision will help readers quickly grasp the scope, structure, and core contributions of the review.
4. The section on the mechanisms of sonodynamic therapy and exosomes is well-written. It is recommended to refer to the expression logic and technical details in the following article to further optimize the clarity and depth of the mechanism description: Nano Research Volume 16, pages 782–791, (2023). This will help to more accurately and comprehensively present the interaction mechanisms between EVs and SDT.
We thank the Reviewer for pointing out this relevant work. Following this suggestion, we have refined the section describing the integration of EVs into SDT by improving the mechanistic clarity and logical flow of the US-EV-tumor interaction. In particular, we now more explicitly describe how US-mediated phenomena (e.g., cavitation, sonoporation) enhance EV uptake, intracellular payload release, ROS generation, and downstream cell death pathways (see page 10, lines 407-418). The article published in Nano Research (2023) has been cited and used as a conceptual reference to improve the chapter “Integration of EVs in SDT”.
5. Some tables in the manuscript (e.g., Table 2) suffer from incomplete content layout and missing information, and the table details should be supplemented and improved. In addition, a small number of sentences have poor grammatical coherence (e.g., the expressions related to "reversing immunosuppression" in the abstract), and the Chinese expression needs to be optimized to enhance the fluency of the text.
We thank the Reviewer for her/his comments regarding the tables in the manuscript. We would like to clarify that all the fitting information from the cited sources has been thoroughly reported. Therefore, any missing parameters were not provided in the original publications (for example, some studies reported only in vivo data, while others included both in vitro and in vivo results). Where possible, we have improved the content and the clarity of the tables while avoiding the addition of excessive information that would impair the efficacy and the readability of the table itself. Specifically, we have restructured the “Engineering strategy”, “Therapeutic Outcome” and “TME Modulation” sections of each table.
Regarding the comment on 'Chinese expression,' we would like to clarify that the manuscript was written entirely in English. However, we have addressed the concerns about fluency and poor grammatical coherence throughout the manuscript, editing both the Abstract and other critical sections to improve readability and precision.
Reviewer 2 Report
Comments and Suggestions for Authors
Authors report on emerged Sonodynamic Therapy as a promising non-invasive approach and innovative strategy that employs extracellular vesicles mostly exosomes as carriers to deliver drugs directly to tumors. Authors also extensively depicted the impact of SDT drug delivery on stimulation of the immune system to fight the disease, paving the way for safer and more effective cancer therapies. Authors also provide very useful info on the increased presence of clinical trials involving the use of SDT as a therapy of brain tumors. Overall, the review is very knowledgeable, comprehensively covering the topic with related references and provide the latest information for researchers dealing with SDT at all aspects.
Author Response
Authors report on emerged Sonodynamic Therapy as a promising non-invasive approach and innovative strategy that employs extracellular vesicles mostly exosomes as carriers to deliver drugs directly to tumors. Authors also extensively depicted the impact of SDT drug delivery on stimulation of the immune system to fight the disease, paving the way for safer and more effective cancer therapies. Authors also provide very useful info on the increased presence of clinical trials involving the use of SDT as a therapy of brain tumors. Overall, the review is very knowledgeable, comprehensively covering the topic with related references and provide the latest information for researchers dealing with SDT at all aspects.
We deeply thank the Reviewer for her/his appreciation and the generous comments. We are glad that our manuscript was found to be comprehensive and informative. We greatly appreciate the recognition of the effort to provide an up-to-date and detailed resource for researchers in this field.